Accepted at the ICLR 2024 Workshop on AI4Differential Equations In Science

# PINA: A PyTorch Framework for Solving Differential Equations by Deep Learning for Research and Production Environments

**Dario Coscia** [*]
Mathematics Area
mathLab, SISSA
via Bonomea 265, I-34136 Trieste, Italy
dario.coscia@sissa.it

**Nicola Demo**
FAST Computing Srl
Via Mazzini 20, 34121, Trieste, Italy
nicola.demo@fastcomputing.net

**Gianluigi Rozza**
Mathematics Area
mathLab, SISSA
via Bonomea 265, I-34136 Trieste, Italy
gianluigi.rozza@sissa.it

## Abstract

We present a versatile software designed for solving differential equations employing neural networks. The package is called PINA, an open-source Python library built upon the robust foundations of PyTorch and Lightning. It allows end-users to formulate their problem and craft their models to effortlessly compute solutions of PDEs by Physics Informed Neural Networks and Neural Operators. The modular structure of PINA permits it to adapt for user specifics, thus offering the freedom to select the most suitable learning techniques for their particular problem domain. Furthermore, by leveraging the capabilities of the Lightning package, PINA adapts to various hardware setups, including GPUs and TPUs. This adaptability positions PINA as an ideal candidate for the transition of these methodologies into production and industrial pipelines, where computational efficiency and scalability are of paramount importance. The package is open-source and available at: https://github.com/mathLab/PINA.

## 1 Introduction

In recent years, the world has seen an unprecedented revolution in artificial intelligence (AI) and machine learning (ML), that has permeated numerous sectors, transforming solutions and processes in many different fields of applied sciences. Within the scientific computing community, this revolution has manifested itself as a powerful tool for overcoming the limitations inherent in traditional methods for solving complex differential equations.

Among the promising developments in this arena, two standout approaches have emerged as central players for differential equation learning: Physics-Informed Neural Networks (PINNs) Raissi et al. (2019) and Neural Operators (NOs) Li et al. (2020); Lu et al. (2021a); Bhattacharya et al. (2021). These methodologies exploit the knowledge of the equations, symmetries, and data to approximate the unknown solution of the differential equation or the differential operator defining the problem. These recent advancements combined with the evolution of open-source frameworks, such as TensorFlow (Abadi et al., 2015), and PyTorch (Paszke et al., 2019) led to the development of several libraries for solving ODEs and PDEs via PINNs and NOs. PINN TensorFlow-based libraries include DeepXDE (Lu et al., 2021b) (which also supports PyTorch), TensorDiffEq (McClenny et al., 2021) and PyDEns (Koryagin et al., 2019); while PyTorch-based libraries include NeuroDiffEq (Chen et al., 2020), IDRLNet (Peng et al.,

---

[*]Corresponding Author

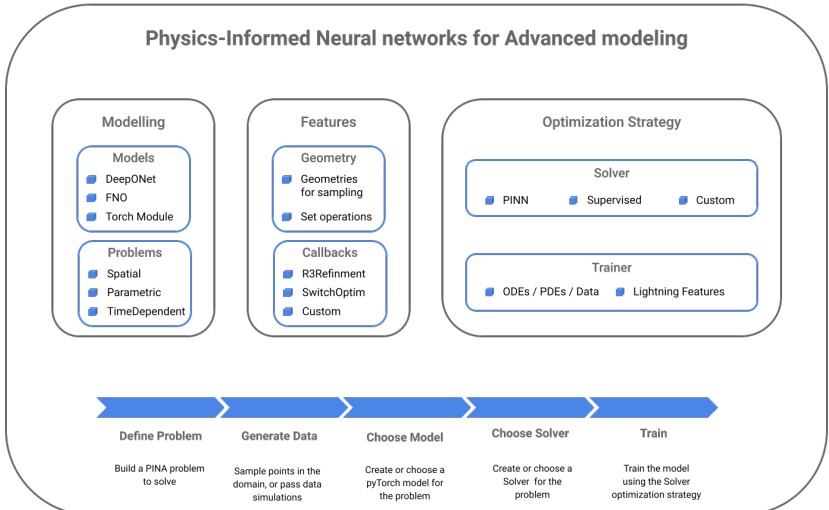

Figure 1: `PINA` package application programming interface. Starting from the problem definition, a specific model is passed to the solver, which defines, together with the trainer, the optimization strategy of the model.

2021). For NO `NeuralOperator` (Li et al., 2020; Kovachki et al., 2021) is the main library. Finally, hybrid software for PDE learning includes (Bouziani & Ham, 2023; Kidger, 2022).

There are multiple challenges with the packages mentioned above that limit their usage for research and production environments. First, most of the packages lack abstract interfaces which limit the possibility of adding extensions, like new loss functions or training procedures. Additionally, the packages presented are sectorized to only PINNs or NOs, without the possibility of combining the two methodologies, which is a new research direction in the field (Li et al., 2021; Wang et al., 2021b). Another common problem of the libraries is the absence of common deep learning advancements for training such as multiple device training, modern model compression techniques, gradient accumulation, and so on. Finally, the possibility of inserting common deep-learning loggers into the training for monitoring is missing. For this reason, we present Physics Informed Neural network for Advanced modeling (`PINA`), an open-source Python library providing an intuitive interface for solving differential equations using PINNs, NOs or both together. The contribution is organized to show features, capabilities, and practical applications of `PINA`, illustrating how this software tool can be exploited for solving complex differential equations using deep learning.

## 2   PINA

Physics Informed Neural network for Advanced modeling (`PINA`) is an open-source Python library built-in `PyTorch`, with `PyTorchLightning` (Falcon & The PyTorch Lightning team, 2019) as backhand to solve differential equations using artificial intelligence model. Employing `PyTorchLightning` as the backhand offers professional AI researchers and machine learning engineers the possibility of using advancement training strategies provided by the library. In addition, it provides the possibility to add arbitrary self-contained routines (callbacks) to the training for easy extensions without the need to touch the underlying code. The application programming interface (API) of `PINA` is schematized in Figure 1. The pipeline to solve differential equations with `PINA` follows five steps: problem definition, data generation, model and solver selection, and training. To show the full capabilities of `PINA` the next sections will follow the prototypical pipeline for solving a problem, highlighting the various features provided by the software. The mathematical notation and a background introduction to PINNs and NOs can be found in Appendix B.

## 2.1 PROBLEM DEFINITION

The first step is the formalization of the problem. In `PINA` the problem is formulated by constructing a class inheriting from one or more problem classes (at the moment the available classes are **AbstractProblem**, **SpatialProblem**, **TimeDependentProblem**, **ParametricProblem**, **InverseProblem**), depending on the nature of the problem treated. For example, a simple ODE that depends only on a spatial variable is defined via a class that inherits only from **SpatialProblem**. Differently, for a parametric time-independent PDE, the problem class inherits from both **SpatialProblem** and **ParametricProblem**. In case the user wants to define its own problem, the **AbstractProblem** interface must be used as the base class. In the problem formulation class, the user must include information about the domains — e.g. spatial, temporal, parametric —, the output variables, and the conditions that the neural network has to satisfy. Multiple types of geometries are available currently in `PINA` for defining the domain (see Section 2.2). The output variables are represented by a list of symbols constituting the unknowns of the problem. Indeed, standard `PyTorch` tensors carry a label (**LabelTensor**) in `PINA`, allowing the user an easy way to manipulate the tensors. Finally, for training PINNs and NOs it is essential to give appropriate constraints as a form of loss function. The **Condition** class encapsulates all the possible ways the loss could be defined, i.e., physical loss, boundary loss, or data loss. The users must use the **Condition** class to define all the constraints the unknown field needs to satisfy. Moreover, `PINA` already implements differential operators (e.g. **laplacian** or **grad**) and common equations (e.g. Dirichlet boundary conditions, systems of equations) to ease the problem formulation for the users.

## 2.2 DATA GENERATION

NO learning procedure uses a finite set of observations and it is trained in a fully supervised manner. These observations, obtained by numerical solver solutions, in `PINA` can be passed as **LabelTensor** in the **Condition** class defined in Section 2.1. Differently, some training strategies, e.g. PINNs, use collocation points sampled inside the domain where the residual of the differential equation (see equation 3) must be evaluated. For these types of solvers, `PINA` provides a simple sampling strategy for multiple different geometries. In `PINA` each domain is a **Location** object, which defines the geometry of the domain. There are already multiple sampling methods in `PINA` e.g. random uniform, grid sampling, or latin hypercube sampling for the different available multidimensional geometries, e.g. hypercube, hypersphere. In addition to multidimensional geometries, the software also provides set operations (difference, union, intersection, and so on) allowing the user to build a custom domain. Finally, in **Condition** class the user can also employ available scatter points, and pass them as **LabelTensor**s.

## 2.3 MODEL AND SOLVER SELECTION

Once the model is defined, the user must choose the neural network model to optimize, and the optimization strategy. In `PINA` the model is represented as a standard **torch.nn.Module**. The package is equipped with many customizable models and layers (see Table 1) already implemented using `PyTorch`. The user can then decide to use built-in models (e.g. for benchmarking) or build new models and layers for research purposes.

Table 1: `PyTorch` models and layers available in `PINA`.

|  | Method | Source |
|---|---|---|
| **Models** | Feed Forward Neural Network (MLP) | - |
|  | Modified MLP | Wang et al. (2021a) |
|  | DeepONet | Lu et al. (2021a) |
|  | MiONet | Jin et al. (2022) |
|  | Fourier Neural Operator (FNO) | Li et al. (2020) |
| **Layers** | Residual Layer | Li et al. (2020) |
|  | Fourier Layer | He et al. (2016) |
|  | Continuous Convolution | Coscia et al. (2023b) |
|  | Spectral Convolution | - |
|  | Proper Orthogonal Decomposition | - |

For optimizing the model a specific solver must be used. A solver is a Python object which defines the optimization strategy for the model. In `PINA` the solver is constructed by inheriting from **SolverInterface**, an abstract class wrapping Lightning Modules. Available solvers include a supervised learning solver (**SupervisedSolver**), particularly crafted for data-driven problems and NO approach,

a physics-informed solver (**PINN**) (Raissi et al., 2019), and an adversarial solver (**GAROM**) (Coscia et al., 2023a). We plan to continuously add solvers as the state–of–the–art evolves. Notice that all solvers are customizable by the user. For example, the **PINN** solver allows changing the loss (e.g. using a variational loss (Kharazmi et al., 2019)), or extending the solver with regularization strategies (Yu et al., 2022), or modifying the optimizer (Davi & Braga-Neto, 2022). All of these, apparently different solvers, can be changed by a keyword argument in the **PINN** class.

## 2.4 PINA TRAINING

The last stage on the `PINA` pipeline consist in training the model. This is done using the **Trainer** class, which wraps the Lightning Trainer class. In the **Trainer** class, the user must pass a **Solver-Interface** object in addition to all the available arguments of the Lightning Trainer. This strategy allows the user maximal training flexibility by exploiting fully `PytorchLightning` capabilities, e.g. low precision training, gradient accumulation, multiple GPU training, and different hardware training. Finally, the **callbacks** argument in the trainer can be used to insert a small part of code at different positions inside the training step.

## 3 EXPERIMENTS

In this section, we show possible benchmark results obtainable with `PINA`. We want to highlight that the purpose of this section is not to provide accurate measurements of model performance, but rather to show how easily is to benchmark on `PINA`. As model cases, we use four different models all implemented in `PINA`: a standard multilayer perceptron (MLP); the skip connection MLP (Wang et al., 2021a) (m-MLP); a hard constraint MLP (Lu et al., 2021c) (hard-MLP); the Deep Operator Network (Lu et al., 2021a) (DeepONet). The models are benchmarked on four different problems using different PINN's learning methodologies: the **Burgers** and **Parametric Poisson** equations, with classical PINN learning; the **Poisson**'s equation using extra features (Demo et al., 2023); and the **Wave** equation, using $R3$ adaptive refinement (Daw et al., 2023). For a complete description of training details and differential problems see Appendix D and C. In Table 2 the mean square residual for all the simulations done employing `PINA` is reported. It is worth mentioning that all simulations have been done by changing just a few lines of code (the problem class, and model definition), which shows the great versatility of the software. Finally, in Figure 2 we show how solutions can be visualized in `PINA` via the software plotting API with the Poisson problem example.

Table 2: Benchmark results for multiple problems and training model in `PINA`.

| Model | Burger | Poisson | Wave | Parametric Poisson |
|---|---|---|---|---|
| MLP | $6.20 \times 10^{-4}$ | $1.87 \times 10^{-7}$ | $1.02 \times 10^{-3}$ | $8.13 \times 10^{-5}$ |
| m-MLP | $4.60 \times 10^{-4}$ | $2.30 \times 10^{-7}$ | $1.71 \times 10^{-4}$ | $6.91 \times 10^{-6}$ |
| hard-MLP | $9.55 \times 10^{-4}$ | $1.67 \times 10^{-6}$ | $4.64 \times 10^{-4}$ | $2.95 \times 10^{-4}$ |
| DeepONet | $2.49 \times 10^{-2}$ | $5.71 \times 10^{-7}$ | $2.02 \times 10^{-2}$ | $5.66 \times 10^{-3}$ |

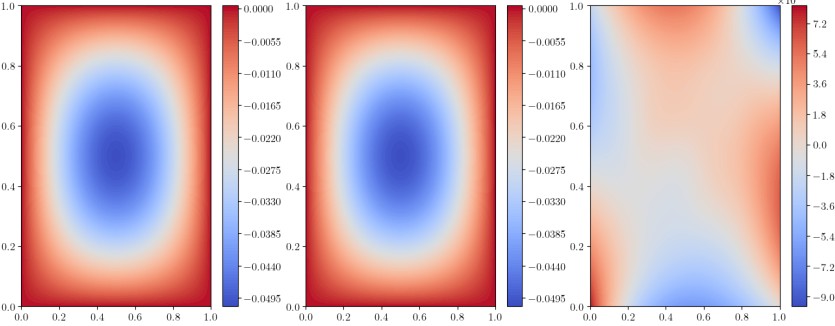

Figure 2: Example of visualization API for the Poisson problem in `PINA`. *Left*: `PINA` solution, *center*: real solution, *right*: absolute value difference of real and predicted solution.

## 4 CONCLUSIONS

We present in this contribution `PINA`, a software framework for learning differential equations leveraging deep learning. With a focus on centralizing research efforts in this domain, `PINA` aims to expedite the integration of these methodologies into production environments while providing a highly customizable entry point for active research. We introduced the most important features, highlighting the modular structure, the `PyTorch` and `PyTorchLighting` inheritance, the extensibility for defining problems and domains, the capability to use several built-in models or crafting from scratch a new one. We showed how `PINA` can be used to solve different problems, using different benchmarking cases.

## ACKNOWLEDGMENTS

The authors thank the reviewers for their time and dedication to providing invaluable comments to improve the manuscript. The authors thank Niccolò Tonicello for his helpful comments on this work. Finally, the authors acknowledge the support from all package contributors. This work is partially supported by European Union Funding for Research and Innovation - Horizon 2020 Program - in the framework of European Research Council Executive Agency: H2020 ERC CoG 2015 AROMA-CFD project 681447 "Advanced Reduced Order Methods with Applications in Computational Fluid Dynamics" P.I. Professor Gianluigi Rozza, by European Union Funding for Research and Innovation — Horizon Europe Program — in the framework of European Research Council Executive Agency: ERC POC 2022 ARGOS project 101069319 "Advanced Reduced order modellinG: Online computational web server for complex parametric Systems" P.I. Professor Gianluigi Rozza, by European High-Performance Computing Joint Undertaking project Eflows4HPC GA N. 955558, by PRIN "Numerical Analysis for Full and Reduced Order Methods for Partial Differential Equations" (NA-FROM-PDEs) project.

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

## A  SOFTWARE

The `PINA` software is available at: https://github.com/mathLab/PINA.

## B  MATHEMATICAL NOTATION AND NEURAL SURROGATE MODELS

ODEs and PDEs are used to describe different physical phenomena in a mathematical form. Local updates expressed by partial or total derivatives are used to represent the evolution of a function characterizing a system. Following the notation presented in (Cuomo et al., 2022), the general form of a differential equation, which we aim to solve, can be written as:

$$
\begin{aligned}
\mathcal{F}(\mathbf{u}(\mathbf{z}); \alpha) &= \mathbf{f}(\mathbf{z}) & \mathbf{z} \in \Omega, \\
\mathcal{B}(\mathbf{u}(\mathbf{z})) &= \mathbf{g}(\mathbf{z}) & \mathbf{z} \in \partial\Omega,
\end{aligned}
\tag{1}
$$

where the solution field is $\mathbf{u}$ living in a suitable space $\mathbb{U}$, the variables $\mathbf{z} = [x_1, \ldots, x_{d_z-1}, t]$ indicate all the spatiotemporal coordinates in a domain $\Omega \subset \mathbb{R}^{d_z}$ with $\partial\Omega$ its boundaries and $d_z$ the space dimension, $\alpha \in \mathbb{A}$ the physical parameters in the suitable space $\mathbb{A} \subset \mathbb{R}^{d_\alpha}$ with $d_\alpha$ the space dimension. Finally, $\mathcal{F}$ is a differential operator describing the dynamics with forcing term $\mathbf{f}$, and $\mathcal{B}$ is the operator which indicates arbitrary initial or boundary conditions, with $\mathbf{g}$ the function on the boundaries.

Solving ODEs and PDEs of the form in Equation equation 1 is one of the main computational challenges in mathematics and engineering. Numerical solvers, such as finite element methods (FEM), finite difference methods (FDM), or finite volume method (FVM), rely on discretizing the domain $\Omega$ (Morton & Mayers, 2005; Quarteroni & Quarteroni, 2009). For many complex domains, the discretization is not straightforward, and a specific study is needed to ensure the final accuracy of the solver. Moreover, these solvers are often computationally expensive, resulting in high energy consumption, and slow computational time.

Over the past decades, multiple deep learning methods have risen for solving the problem formalized in equation 1, aiming to overcome the classical numerical solver issues. Eventually, a dichotomy of methodologies can be made: Neural Operator (NO) methods, which assume knowledge of the system in the form of data; and Physics Informed Neural Networks (PINNs), which use the underlying equation to learn the solution.

### B.1  NEURAL OPERATOR METHODS

Neural Operator (NO) methods (Li et al., 2020; Lu et al., 2021a; Bhattacharya et al., 2021; Kovachki et al., 2021; Brandstetter et al., 2022) build a mapping from infinite-dimensional function spaces by using a supervised learning strategy. Given a specific ODE or PDE as in the form of equation 1, a neural operator $G : \mathbb{U}' \to \mathbb{U}$ is trained by a finite set of $N$ observations $\{(\mathbf{u}'_i, \mathbf{u}_i)\}_{i=1}^N$, such that:

$$
G(\mathbf{u}'_i) \approx \mathbf{u}_i \quad \forall i = 1, \ldots, N.
\tag{2}
$$

For example, a NO could map the field at the initial temporal condition of a PDE, to the evolution at a specific time step; or the parameter of a differential equation to its solution for the specific parameter. NO possesses important characteristics: they are discretization invariant, i.e. the model is not defined on a fixed grid; the input and output is a function; the universal approximation theorem for operator holds Chen & Chen (1995).

### B.2  PHYSICS INFORMED NEURAL NETWORKS

In many situations training data are not available, and NO can not be trained using a supervised loss. As an alternative, PINNs (Raissi et al., 2019) have been proposed. PINNs are trained by approximating the true solution of equation 1 with a neural network $\mathbf{u}_\theta \approx \mathbf{u}$ with parameters $\theta$. In PINNs the network is trained directly with the ODE or PDE itself, ensuring that equation 1 is satisfied by the network:

$$
\mathcal{L}(\theta) = \mathcal{L}_\mathcal{F} + \mathcal{L}_\mathcal{B}.
\tag{3}
$$

The first term is the *physics-informed* loss inside the domain $\Omega$, while the second one is a supervised loss for boundary or initial conditions. Different types of losses can be implemented, for example

using the MSE loss equation 3 becomes:

$$\mathcal{L}(\theta) = \frac{1}{N_f} \sum_{i=1}^{N_f} \|\mathcal{F}(\mathbf{u}_\theta(\mathbf{z}_i); \alpha_i) - \mathbf{f}(\mathbf{z}_i)\|_2^2 + \frac{1}{N_b} \sum_{i=1}^{N_b} \|\mathcal{B}(\mathbf{u}_\theta(\mathbf{z}_i)) - \mathbf{g}(\mathbf{z}_i)\|_2^2, \qquad (4)$$

where $N_f$ is the number of collocation points sampled inside $\Omega$, and $N_b$ the number of collocation points sampled in $\partial\Omega$.

Since PINN's inception, many follow-up improvements have been made to improve training stability and convergence. Examples of further research include studying different losses (Kharazmi et al., 2019; Wang et al., 2022; McClenny & Braga-Neto, 2020), sample strategies for collocation points (Wu et al., 2023; Nabian et al., 2021; Daw et al., 2023) to speed up convergence, or specific network architecture (Wang et al., 2021a;b) and input augmentation (Demo et al., 2023; Lu et al., 2021c) to ease the neural network training.

## C    DIFFERENTIAL PROBLEMS

In this section, we provide the mathematical formulations of the problem presented in the experiment section 3.

### C.1    BURGER'S EQUATION

Burger's equation is a convection-diffusion equation widely used in many fields of mathematics. The problem is crafted as the benchmark presented in (Raissi et al., 2019). Let $\boldsymbol{x} = (x, t)$ be the spatio-temporal variables, and $u$ be the unknown field. The Burger equation is:

$$\begin{cases} \frac{\partial}{\partial t} u(\boldsymbol{x}) + u(\boldsymbol{x}) \frac{\partial}{\partial x} u(\boldsymbol{x}) - \frac{0.01}{\pi} \frac{\partial^2}{\partial x^2} u(\boldsymbol{x}) = 0 & x \in [-1, 1], t \in [0, 1] \\ u(1, t) = u(-1, t) = 0 & t \in [0, 1] \\ u(x, 0) = -\sin(\pi x) & x \in [-1, 1]. \end{cases} \qquad (5)$$

For solving the equation we sample 10000 points uniformly random in the domain $[-1, 1] \times [0, 1]$.

### C.2    POISSON'S EQUATION

Poisson's equation is an elliptic partial differential equation widely used in physics. The problem is crafted as the benchmark presented in (Demo et al., 2023). Let $\boldsymbol{x} = (x, y)$ be the spatial variables, $u$ be the unknown field, and $\Omega = [-1, 1]^2$ the domain. The Poisson equation is:

$$\begin{cases} \nabla^2 u(\boldsymbol{x}) = \sin(\pi x) \sin(\pi y) & \boldsymbol{x} \in \Omega \\ u(\boldsymbol{x}) = 0 & \boldsymbol{x} \in \partial\Omega, \end{cases} \qquad (6)$$

where $\partial\Omega$ indicates the boundary of the domain, and the Laplacian operator $\nabla^2$ acts on the spatial variables. For solving the equation we sample 10000 points uniformly random in the domain $\Omega$. During problem learning we employ extra features, by augmenting the input with the forcing term, i.e. the model input is given by $(x, y, \sin(\pi x) \sin(\pi y))$.

### C.3    WAVE'S EQUATION

The Wave's Equation is a linear differential equation vastly used in fluid dynamics. Let $\boldsymbol{x} = (x, y, t)$ be the spatio-temporal variables, $u$ be the unknown field, $\Omega = [0, 1]^2$ the domain, and $\mathbb{T} = [0, 1]^2$ the parameter domain. The Wave equation is:

$$\begin{cases} \nabla^2 u(\boldsymbol{x}) = \frac{\partial^2}{\partial t^2} u(\boldsymbol{x}) & \boldsymbol{x} \in \Omega \times \mathbb{T} \\ u(\boldsymbol{x}) = 0 & \boldsymbol{x} \in \partial\Omega \times \mathbb{T}, \\ u(\boldsymbol{x}) = \sin(\pi x) \sin(\pi y) & \boldsymbol{x} \in \Omega \times \partial\mathbb{T}, \end{cases} \qquad (7)$$

where $\partial\Omega$ indicates the boundary of the domain, and the Laplacian operator $\nabla^2$ acts on the spatial variables. For solving the equation we sample 10000 points uniformly random in the domain $\Omega$. We use $R3$ adaptive refinement for moving the collocation points during training every 100 epochs.

## C.4 Parametric Poisson's equation

Parametric Poisson's equation is an example of a Poisson equation where the forcing term depends on external parameters. The problem is crafted as the benchmark presented in (Demo et al., 2023), where the objective is to learn a function for different parameters. The problem can be considered as a NO problem since we map different initial functions (for different parameters) to the field functions. In the experiment section, we use PINN learning to solve the problem. Let $\boldsymbol{x} = (x, y)$ be the spatial variables, $u$ be the unknown field, $\Omega = [0,1]^2$ the domain, and $\Xi = [-1,1]^2$ the parameter domain. The Poisson equation is:

$$\begin{cases} \nabla^2 u(\boldsymbol{x}) = e^{-2[(x-\xi_1)^2 + (y-\xi_2)^2]} & \boldsymbol{x} \in \Omega \times \Xi \\ u(\boldsymbol{x}) = 0 & \boldsymbol{x} \in \partial\Omega \times \Xi, \end{cases} \tag{8}$$

where $\partial\Omega$ indicates the boundary of the domain, and the Laplacian operator $\nabla^2$ acts on the spatial variables. For solving the equation we sample 10000 points uniformly random in the domain $\Omega$.

## D  Experiment Details

In this section, we provide the network specifics for the experiments performed in Section 3. All the models were trained using the Adam optimizer (Kingma & Ba, 2014), with a learning rate of 0.001 for 10000 epochs minimizing the mean square error loss. The training was done on an Intel CPU.

### D.1  Burger's equation

The networks' composition:

- **MLP**: Three linear layers of size $[20, 10, 5]$ with hyperbolic tangent activation on all layers except the last

- **m-MLP**: Three linear layers of size $[20, 20, 20]$ with hyperbolic tangent activation on all layers except the last. The transformer networks were two linear layers mapping the input to the inner size of 20

- **hard-MLP**: Same as **MLP**. Hard constraints on boundary conditions are imposed by multiplying the network output with the term $(1 + x)(1 - x)$

- **DeepONet**: The branch and trunk net are the same architecture of two linear layers of size $[20, 20]$ with hyperbolic tangent activation on all layers except the last. The reduction is done by aggregating with a linear layer with input dimension 20 and output dimension 1. The trunk net takes $t$ as input. The branch net takes $x$ as input.

The input dimension of the problem is 2 (one spatial + one temporal variables), and the output dimension is 1.

### D.2  Poisson's equation

The networks' composition:

- **MLP**: Same architecture as Burgers's problem.

- **m-MLP**: Same architecture as Burgers's problem.

- **hard-MLP**: Same architecture as Burgers's problem.

- **DeepONet**: Same architecture as Burgers's problem, but the trunk net takes $x, t$ as input. The branch net takes $\sin(\pi x)\sin(\pi y)$ as input.

The input dimension of the problem is 3 (two spatial + augmentation variables), and the output dimension is 1.

### D.3 WAVE'S EQUATION

The networks' composition:

- **MLP**: Same architecture as Burgers's problem.
- **m-MLP**: Same architecture as Burgers's problem.
- **hard-MLP**: Same architecture as Burgers's problem.
- **DeepONet**: Same architecture as Burgers's problem, but the trunk net takes $t$ as input. The branch net takes $x, y$ as input.

The input dimension of the problem is 3 (two spatial + two parametric variables), and the output dimension is 1.

### D.4 PARAMETRIC POISSON'S EQUATION

The networks' composition:

- **MLP**: Same architecture as Burgers's problem.
- **m-MLP**: Same architecture as Burgers's problem.
- **hard-MLP**: Same architecture as Burgers's problem.
- **DeepONet**: Same architecture as Burgers's problem, but the trunk net takes $x, t$ as input. The branch net takes $\xi_1, \xi_2$ as input.

The input dimension of the problem is 4 (two spatial + two parametric variables), and the output dimension is 1.

