# OpenReview forum: "PINA: a PyTorch Framework for Solving Differential Equations by Deep Learning for Research and Production Environments"
_ICLR.cc/2024/Workshop/AI4DiffEqtnsInSci — AI4DiffEqtnsInSci @ ICLR 2024 Poster_

### Official Review · Reviewer_dDbJ · 2024-02-15
**paper introducing a pytorch-based open-source package for solving PDEs and operator learning**

**Rating:** 3
**Confidence:** 5

**Review:**

The paper introduces PINA (Physics Informed Neural network for Advanced modeling) which is a PyTorch-based open-source package for solving PDEs and operator learning via neural networks. As argued in the paper, the main advantage of PINA compared to existing packages is that it streamlines the training process by adding interfaces for logging or developing new loss functions.

I would like to congratulate the authors for their efforts in building an open-source package. However, in my professional opinion, the technical contributions of this work are marginal. Specifically, while having a streamlined pipeline is useful, the already available libraries make it very easy to implement any of the models that the authors have exemplified in Sec. 4. The performance of this package (and many of its claimed benefits) are also not compared to existing open-source packages so it is difficult to evaluate the authors’ claims.

---

### Official Review · Reviewer_xssh · 2024-02-28
**A PyTorch Lightning's wrapper for PINNs and/or NOs**

**Rating:** 6
**Confidence:** 4

**Review:**

This paper introduces a new PyTorch interface, PINA, for developing ML models based on the PINN and/or NO methodologies. PINA is built upon Lightning, and therefore benefits from its features. This is achieved mainly by providing a Trainer class wrapping Lightning’s trainer. Details are provided about the API, and a few results are shown.

The proposed framework seems useful for implementing various existing ML methods for solving PDE-based problems. The engineering effort behind PINA provides a productive way to develop new methods and can facilitate the deployment of those methods on different hardware setups.

While the engineering effort deployed to compose the interface with Lightning can be useful, the originality/novelty of the proposed framework is questionable and there are a couple of flaws in the mathematical formulation of the problem (see below). More specifically, the main contribution is to provide a wrapper around Lightning’s Trainer class and most of the claimed advantages of PINA over related methods in the literature (e.g., multiple device training, modern model compression techniques, gradient accumulation, logger insertion for monitoring, etc.) are consequences of having a Lightning backend and do not result from new features introduced in this work. Also, the paper mentions that existing packages can’t combine the PINN and NO methodologies. However, the DeepXDE package [1] provides support for PINNs as well as DeepONet, a well-known type of NO. In addition, further experiments comparing PINA against competitors from the literature could improve the dissemination of the package.

As for the related works, the paper misses out on prior works on the subject of software for PDE learning, see [2, 3], which should be included.

Lastly, the paper contains several flaws in the mathematical formulation of the problem:

- There are a few problems with equation 1:
	- z is said to indicate the spatio-temporal coordinates, however $z \in \Omega$, where $\Omega$ is the spatial domain. In fact, equation 1 is effectively a steady problem, i.e. with no time dependency, which is a pretty restrictive assumption for software tackling PDEs.
	- $\mathscr{B}$ cannot be used for initial conditions given that the problem is time-independent.
	- $d_{z}$ is not defined
	- $\Omega \in \mathbb{R}^{d_{z}}$ → $\Omega \subset \mathbb{R}^{d_{z}}$
	- the spaces $A$ and $U$ are not defined

- In equation 2, $u^{\prime}$ and $u$ are not defined. I assume you meant $G(u\prime_{i}) = u_{i} \quad \forall 1 \le i \le N$ ?


[1] Lu et. al, DeepXDE: A deep learning library for solving differential equations, SIAM Review, arXiv.1907.04502, 2021.

[2] N. Bouziani and D. A. Ham, Physics-driven machine learning models coupling PyTorch and Firedrake, ICLR 2023 Workshop on Physics for Machine Learning, arXiv.2303.06871, 2023.

[3] P. Kidger, Diffrax - On Neural Differential Equations, University of Oxford, arXiv:2202.02435, 2021.

---

### Meta-Review · Area_Chair_9XnF · 2024-03-01

**Recommendation:** Accept (Poster)

**Metareview:**

This paper introduces a new PyTorch interface, PINA, for developing ML models based on the PINN and/or NO methodologies. PINA is built upon Lightning, and therefore benefits from its features. This is achieved mainly by providing a Trainer class wrapping Lightning’s trainer. The contribution is positive for the community but the performance of this package is not compared to existing open-source packages and there is limited scientific advances made in this work so with this in mind it will be accepted but only for a poster given some of the limitations of this paper.

---

### Decision · Program_Chairs · 2024-03-01

Accept (Poster)